# [Re] Understanding Self-Supervised Learning Dynamics without Contrastive Pairs

## Reproducibility Summary

**Scope of Reproducibility**

The authors in [1] claim that with the underlying learning dynamics of *BYOL* [2] and *SimSiam* [3], a new method *DirectPred* can be derived. We investigate the assumptions made for this derivation and also compare the quality of the produced encoder representations through linear probing of these networks.

**Methodology**

We reimplemented *BYOL*, *SimSiam* and *DirectPred* from scratch as well as their ablations in TensorFlow. We checked original repository in written PyTorch for some implementation details. In all experiments we used the CIFAR-10 train set for training and the test set for evaluation. We were running our experiments for more than 100 hours on GCP's V100 GPU.

**Results**

We show that the theoretical assumption regarding eigenspace alignment and symmetry hold also for a different dataset other than the one used in the original paper. In addition, we reproduce ablations regarding learning rate, weight decay and Exponential Moving Average.

Since we used CIFAR-10 in all experiments we can not directly compare accuracies. However, we show the same relative behaviour of different networks given hyperparameter changes. We can directly compare performance for one of the experiments (Table 8. in [1] bottom left part). Our models, namely *SGD Baseline*, *DirectPred* (with and without frequency=5), achieve comparable accuracy which differ by at most 1%. We also confirm the claim that *DirectPred* outperforms its one-layer SGD alternative. Our code can be accessed under the following link: `https://anonymous.4open.science/r/SelfSupervisedLearning-FDOF`.

**What was easy**

The architecture of the Siamese network and training schemes were both straightforward to implement and easy to understand.

**What was difficult**

We could not run our code on STL-10 dataset due to time and resource constraints. Due to differences between PyTorch and TensorFlow libraries, we had to implement some parts by hand to keep our code as close to the original work as possible. Also, original repository is not easy to read and does not cover all the experiments (e.g. eigenspace alignment experiment). Correctly applying data-augmentation was also a hard task due to assumptions of how the individual data augmentations functions actually work.

## Communication with original authors

We did not contact authors of the paper since we did not encounter any major issues during the reproducibility study.

## 1   Introduction

Self-Supervised learning has become an important task in many domains, since labeled data is often rare and expensive to get. Many modern methods of Self-Supervised learning are based on Siamese-networks [4] which are weight sharing Neural networks for two or more inputs which representations then will be compared in latent space. The representation created by this approach can then be used for classification by fine-tuning on fewer labelled data-points. Traditionally, during pre-training positive pairs (same image, or two images from the same class) and negative pairs (different images or two images from a different class) are used. The distance of the representation of positive pairs is minimized while the distance of the representation of negative pairs is maximized, which prevents the networks from collapse (i.e mapping all inputs to the same representation). These methods have shown quite some success in the past [5], [6], [7], [8]. However, these methods rely on negative pairs, and large batch sizes which makes the training less feasible.

Recently, new methods have been proposed which rely only on positive pairs and yet don't collapse [2], [3]. In the paper "Understanding Self-Supervised Learning Dynamics without Contrastive Pairs" by Tian et.al. [1] the underlying dynamics are explored and based on the theoretical results, a new method, *DirectPred*, was proposed which does not need an update of the predictor via gradient descent but instead is set directly each iteration.

The focus of this work is to test several assumptions made in [1] for the theoretical analysis and see if they hold. For this, we will concentrate especially on the eigenvalues of the predictor network and the eigenspace alignment with its input. Also, we will reproduce the results from [1], [2] and [3] on CIFAR-10 to compare their learned representation using linear probing.

## 2   Related work

A common approach to representation learning without Siamese networks is generative modelling. Typically these methods model a distribution over the data and a latent space, from which then embeddings can be drawn as data representations. Usually these approaches rely on Auto-encoding [9, 10] or Adversarial networks [11, 12]. However, generative models are often computatinaly heavy and hard to train.

Discriminative methods using Siamese networks like SimCLR [5, 6] and Moco [7] outperform generative models and have lower computational cost. However, these methods rely on very large batch sizes since they use contrastive pairs. Most recent methods, replicated in this work, like *BYOL* [2] and *SimSiam* [3], only rely on positive pairs and therefore can make use of smaller batch sizes. To understand why these methods do not collapse, the dynamics of these networks are analysed with linear models in [1, 13]. From this analysis, the authors could derive ablations of *BYOL* where part of the network is directly set to its optimal solution instead of being trained by gradient descent.

## 3   Method

In this section we will describe the methods of *BYOL* and *SimSiam* as well as their successor *DirectPred*.

### 3.1   BYOL & SimSiam

The network architecture of the models is shown in Figure 1. First, two augmented views $X_1'$ and $X_2'$ of an image $X$ are created and fed into the online network $W$ and target network $W_a$ respectively. Both of these networks have the same architecture, a ResNet-18 ($W_{enc}^x$) as encoder [14], which is supposed to create hidden features and a projector head $W_{pro}^x$, which is a two layer MLP, with purpose to map the feature space into a lower dimensional hidden space. The online network also has an additional predictor head, again consisting of a two layer MLP. The target network has a *StopGrad* function instead of a predictor head. Therefore during back propagation, only the weights of the online network are updated via gradient decent. The loss between the output of the online and target network is equal to the cosine-similarity loss function.

$$\mathcal{L}(\hat{Z}_1^{(O)}, \hat{Z}_2^{(T)}) = -\frac{\langle Z_1', Z_2' \rangle}{||Z_1'||_2 ||Z_2'||_2} \tag{1}$$

Note, that the final loss of one image is the symmetric loss $\mathcal{L}(\hat{Z}_1^{(O)}, \hat{Z}_2^{(T)}) + \mathcal{L}(\hat{Z}_2^{(O)}, \hat{Z}_1^{(T)})$, since each augmentation is given to both networks. As mentioned, the target network is not updated with gradient descent, but with an exponential

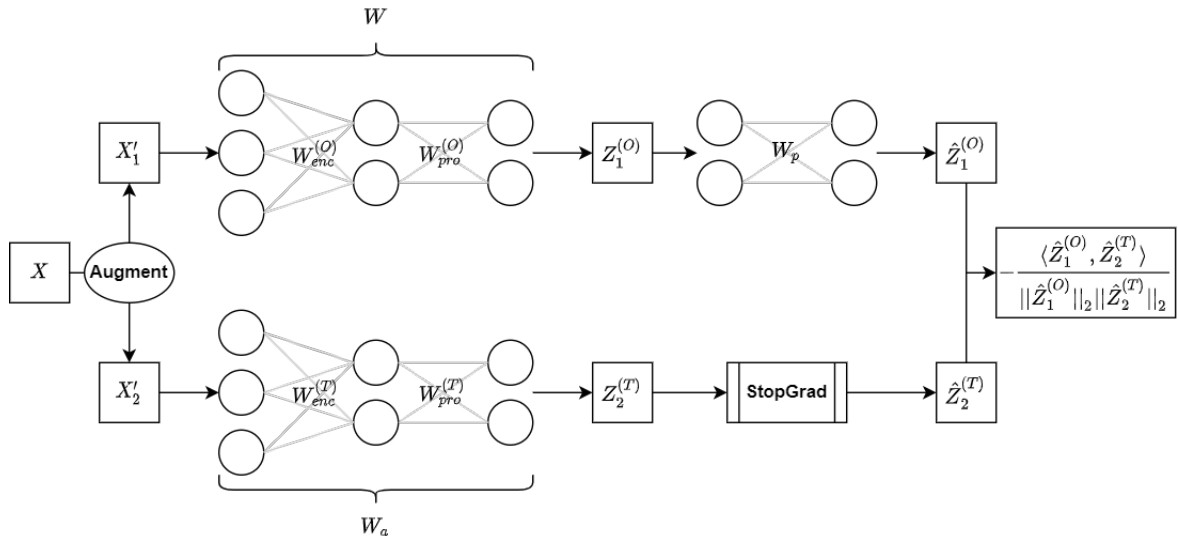

Figure 1: Network architecture for all presented methods

moving average (EMA). After each batch the target network will be set to $W_a = W_a + (1 - \tau)(W - W_a)$. In *SimSiam* the target network is set directly to the online network after each update, i.e $\tau = 0$.

## 3.2 DirectPred

[1] derives a one layer predictor head analytically with the analysis of the underlying learning dynamics of the models presented in Section 3.1 with an approximation of the actual network as a purely linear model. The learning dynamics of the networks are

$$\dot{W}_p = \alpha_p(-W_pW(X + X') + W_aX)W^\top - \eta W_p \tag{2}$$

$$\dot{W} = W_p^T(-W_pW(X + X') + W_aX) - \eta W \tag{3}$$

$$\dot{W}_a = \beta(-W_a + W) \tag{4}$$

With $X = \mathbb{E}[\hat{x}\hat{x}^\top]$, where $\hat{x}$ is the average augmented view of a datapoint and $X'$ is the covariance matrix of the augmented views. $\alpha_p$ and $\beta$ are multiplicative learning rate ratios, i.e $\alpha_p = \frac{\alpha_{\text{pred}}}{\alpha}$ and $\beta = \frac{1-\tau}{\alpha}$ (here $\alpha$ and $\alpha_{\text{pred}}$ are the learning rates for $W$ and $W_p$ respetively). In addition to the linearity of the network, three simplifying assumptions where made:

- The target network is always in a linear relationship with the online network (e.g. $W_a(t) = \tau(t)W(t)$

- The original data distribution $p(X)$ is Isotropic and its augmentation $\hat{p}(X'|X)$ has mean $X$ and covariance $\sigma\mathbf{I}$

- The predictor $W_p$ is symmetric

Based on these assumptions, one can show, that the eigenspaces of the output of the online network and the predictor $W_p$ align. Let $F = WXW^\top$ (i.e. the output of the online network when it is approximated as a linear model), then it follows with the three assumptions, that the eigenspaces of these two matrices align over time (e.g. for all non-zero eigenvalues $\lambda_{W_p}, \lambda_F$ of $W_p$ and $F$, the corresponding normalized eigenvectors $v_{W_p}, v_F$ are parallel, $v_{W_p}^\top v_F = 1$). With this alignment one can derive decoupled dynamics for the eigenvalues of $W$ and $W_p$. By analysing this system, it can be shown that it has, depending on the weight decay parameter, several fixpoints, from which some are stable and some not. The trivial solution (the collapse) is one of them and the basin of attraction of these fixpoints varies with the relative learning rate of the predictor $\frac{\alpha_{\text{pred}}}{\alpha}$. With this analysis, [1] derives conditions under which the trivial fixpoint can be avoided. For a thorough mathematical analysis, we refer to [1]. In Section 5.1 we will present empirical evidence, that the symmetry assumption holds, and that the eignenspaces align. Furthermore, in Section 5.3 we will investigate the role of weight decay and the learning rate.

From the decoupled dynamics of the eigenvalues, we can also derive an analytical expression for the predictor $W_p$. Let $F = U\Omega U^\top$ be the eigen-decomposition of $F$ with $\Omega = \text{diag}(\lambda_F^{(1)}, ..., \lambda_F^{(d)})$ the diagonal matrix with the eigenvalues of $F$, then we can approximate the eigenvalues of $W_p$ with

$$\lambda_{W_p}^{(j)} = \sqrt{\lambda_F^{(j)} + \epsilon \max_j \lambda_F^{(j)}} \tag{5}$$

and therefore set $W_p$ to

$$W_p = U\text{diag}(\lambda_{W_p}^{(1)}, ..., \lambda_{W_p}^{(d)})U^\top \tag{6}$$

Note, that we cannot compute $F$ directly, which is why we use a running average $\hat{F}$ as approximation in practice

$$\hat{F} = \rho\hat{F} + (1 - \rho)\hat{Z} \tag{7}$$

where $\hat{Z} = \hat{Z}_1^{(O)}\hat{Z}_2^{(O)\top}$.

We denote this method *DirectPred* and in Section 5.2 we show, that *DirectPred* can perform similar to *BYOL* and *SimSiam*

# 4    Data & Configurations

We ran our experiments on Google Cloud Platform using Virtual Machine with a V100 GPU.

All experiments are conducted on CIFAR-10 [15], which contains 60 000 RGB images uniformly distributed over 10 classes. The pre-training and the linear evaluation are done on the entire training set, which consists of 50 000 images. For the linear evaluation, only a linear layer is used on top of the encoder, where the weights of the encoder are frozen (i.e. we test how linearly separable the encoders output is). The reported accuracy results are produced from a test set containing 10 000 images. Also, to account for the small dimension of the CIFAR-10 images ($32 \times 32 \times 3$) we use $3 \times 3$ convolutions and stride 1 without maximum pooling in the first block of the encoder.

To augment each image, we first do a random flip, take a random crop (up to 8% of the original size) of the image. Then we randomly adjust brightness, saturation, contrast and hue of the RGB image by a random factor [1]. Finally with a 20% chance we convert the image to grey scale.

**Self-supervised pretraining**    In the basic setting, the online network use ResNet-18 as encoder, two layer projector MLP, two layer predictor MLP, where the first layer consists of 512 nodes, followed by BatchNorm and ReLU, and then a linear output layer with 128 nodes. For *BYOL* we use EMA to update target network and for *SimSiam* we directly set encoder and projector of target network to the weights of the online one ($\tau = 0$). We use SGD optimizer with learning rate 0.03, momentum 0.9 and weight decay (L2 penalty) of 0.0004. The predictor of *DirectPred* is set directly and are not trained with gradient descent and consist of one linear layer with 128 nodes. By SGD baseline for those methods we mean a network pre-trained with a one linear layer predictor with or without EMA. In all experiments, we use batch size of 128. For updating the target network we used the EMA parameter $\tau = 0.996$. For *DirectPred* we use $\epsilon = 0.1$ and $\rho = 0.3$.

**Linear evaluation**    In order to test the performance of the different models, we use linear evaluation, i.e. we train a linear layer on top of the ResNet-18 encoder with frozen weights for 100 epochs. This measures how linearly separable the learned representations of the encoder are. We use Adam optimizer [16] with polynomial decay of learning rate from 5e-2 to 5e-4. Images are normalized but we do not use augmentation for this part of training just as in the original repository for *DirectPred*.

# 5    Experiments and findings

In this section, we will first show that the assumptions and theoretical findings from Section 3.2 hold in practice. Finally, we will pre-train and use linear evaluation on the different models presented in Section 3 in order to test their performances.

---

[1]for brightness, saturation and contrast we chose a value uniformly at random between 0.6 and 1.4. For adjusting the hue, we set the maximal value to 0.1

## 5.1 Eigenspace alignment

First, we pre-train *BYOL* and *SimSiam* keep track of the predictor heads symmetry and eigenspace alignment. In Figure 2 we can see, that the assumption of an symmetric predictor $W_p$ holds. Even without symmetry regularisation, $W_p$ approaches symmetry during training. Also, we can see that for all non-zero eigenvalues of $W_p$ the eigenspaces between $F$ and $W_p$ align as the training progresses.

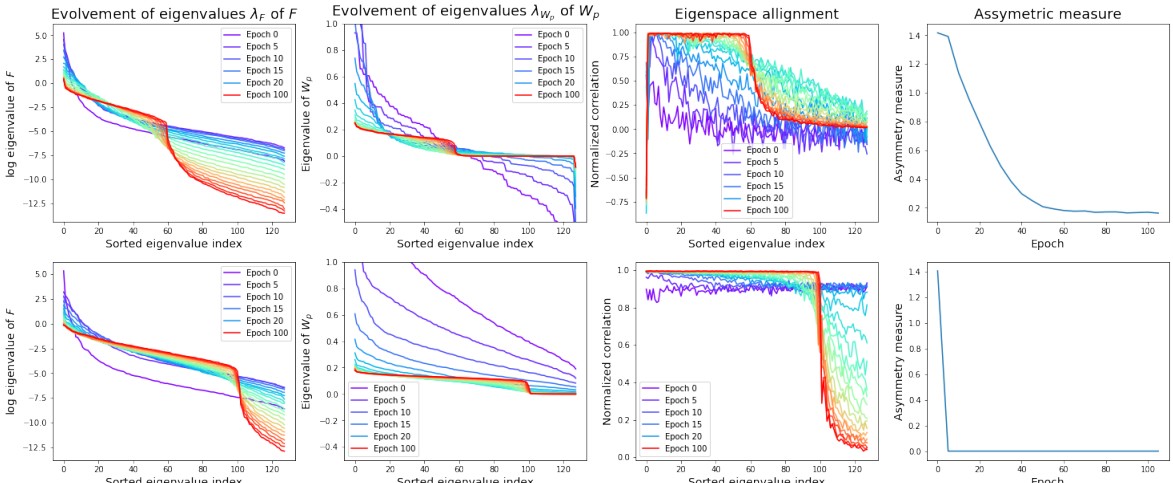

Figure 2: Pre-training *BYOL* for 100 epochs of CIFAR-10. **Top row:** *BYOL* without symmetry regularisation on $W_p$. **Bottom row:** *BYOL* with symmetry regularisation on $W_p$. The eigenvalues of $F$ are plotted on the log scale, since the eigenvalues vary a lot. The assymmetry is measured by $\frac{||W_p - W_p^\top||}{||W_p||}$

We ran the same Experiment for *SimSiam*, and can also see the same effect on the predictor and the alignment (Figure 3). If we don't use a symmetric predictor, we also see that the eigenspaces for the non-zero eigenvalues align. However, once we use symmetry regularisation on $W_p$, all eigenvalues become zero, which shows that the network collapses. We will see later in Section 5.3 that we can prevent this collapse by using different learning rates $\alpha, \alpha_{\text{pred}}$ and weight decay $\eta, \eta_{\text{pred}}$ for $W$ and $W_p$ respectively.

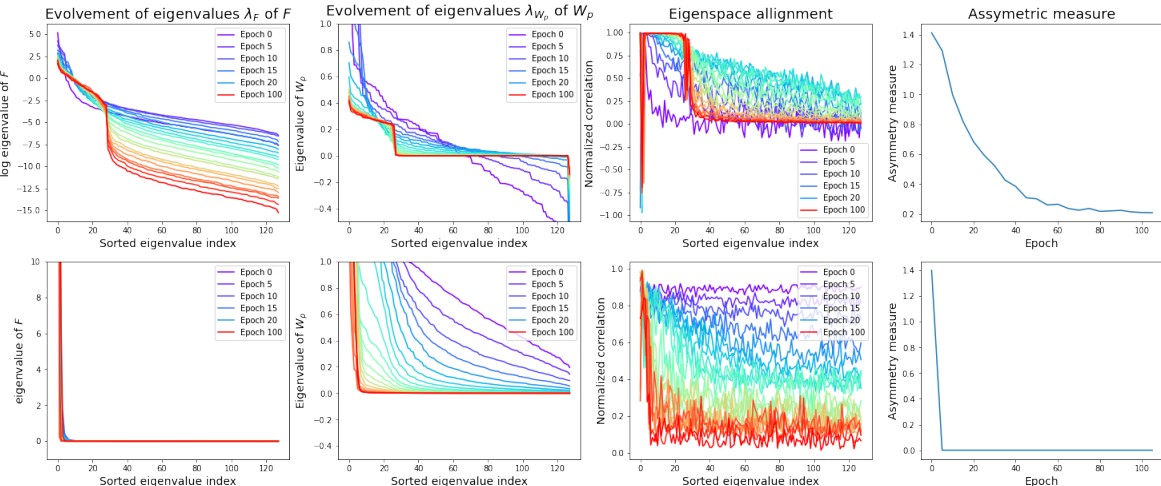

Figure 3: Pre-training *SimSiam* for 100 epochs of CIFAR-10. **Top row:** *SimSiam* without symmetry regularisation on $W_p$. **Bottom row:** *SimSiam* with symmetry regularisation on $W_p$. Note that the eigenvalues of $F$ are not plotted on the log scale here, since we get 0 values.

## 5.2   Performance

**Byol & SimSiam**   In table 1 we can see that the performance of *BYOL* increases slightly when using symmetry regularisation on the predictor. However, as already seen in Figure 3, when using no EMA, we observe that the network collapses. We observe in general better performance for models trained with EMA, given the same hyperaramers. However, we did not use extensive hyperparameter tuning, as performance is not the focus of our work.

|        | symmetric $W_p$ | non symmetric $W_p$ |
|--------|:---------------:|:-------------------:|
| EMA    | **85.7**        | 84.2                |
| No EMA | 20.3            | **79.4**            |

Table 1: Comparision of a two layer predictor with and without symmetry regularisation as well as with and without EMA (i.e first row is *BYOL* and second row is *SimSiam*).

**DirectPred**   As we can see in Figure 2 & 3, the eigenspaces for both models align and therefore the theoretical assumptions of [1] hold. As we can see in Table 2, all models perform reasonably well, and can achieve almost the same performance as *BYOL* or *SimSiam*. However, as already mentioned earlier, we can see that models with EMA outperform models without EMA. I addition, we run an experiments where the predictor is only updated every 5th step according to Equation 6 and otherwise is updated with gradient decent, we call this method *DirectPred$_5$*. We can see that the hybrid method *DirectPred$_5$* does not increase performance, however, according to [1] when training for 500 epochs, *DirectPred$_5$* can outperform *DirectPred*. Due to computational constraints we cannot reproduce this experiment.

|        | SGD Baseline | DirectPred | DirectPred$_5$ |
|--------|:------------:|:----------:|:--------------:|
| EMA    | 83.3%        | **84.7%**  | 84.1%          |
| No EMA | 77.8%        | 78.6%      | -              |

Table 2: The accuracies of SGD baselines, DirectPred and DirectPred with Frequency 5 with and without EMA

## 5.3   Influence of weight decay and learning rate

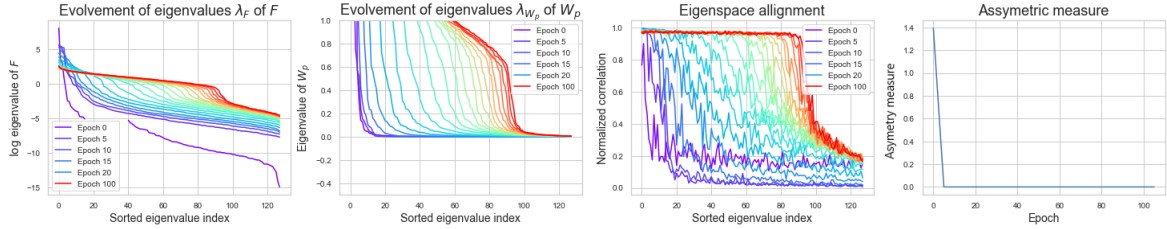

Figure 4: *SimSiam* with symmetric predictor but learning rates $\alpha = 0.2$, $\alpha_{\text{pred}} = 2$ and weight decay $\eta = 0$, $\eta_{\text{pred}} = 4e - 4$

As we can see in Figure 3, *SimSiam* with symmetric predictor does collapse. However, we can prevent this by adjusting the weight decay and learning rate. To make sure the network does converge to a stable non-collapsing fix-point, the weight decay of the predictor should be set higher than the rest of the network ($\eta_{\text{pred}} > \eta$, for mathematical analysis see [1]). By omitting weight decay, we are not able to stabilize the training of *SimSiam* with symmetric predictor and we can also see, that methods without weight decay perform worse, than with weight decay (Table 3). Also, to decrease the basin of attraction, of the trivial fixpoints, the learning rate of the predictor should be rather large compared to the learning rate of the rest of the network, i.e $\frac{\alpha_{\text{pred}}}{\alpha} >> 1$ (see Section 3.2 in [1]).

## 6   Challenges

The original paper describes the methods and mathematical derivations well. Authors also share which hyperparameters they used in most of the experiments. Since the authors provided the open-source repository for the paper, we could

| | symmetric $W_p$ | regular $W_p$ |
|---|---|---|
| $\eta = 0$ & $\eta_{\text{pred}} = 0.0004$ | | |
| Byol | 81.34 % | 81.69 % |
| SimSiam | 79.1 % | 81.39 % |
| $\eta = \eta_{\text{pred}} = 0$ | | |
| Byol | 80.78 % | 80.42 % |
| SimSiam | 20.27 % | 79.22 % |

Table 3: Byol and SimSiam trained with different values for the weight decay parameters. For all experiments in this Table, we set the learning rates $\alpha_{\text{pred}} = 2$ and $\alpha = 0.2$. Note, that the important condition is $\frac{\alpha_{\text{pred}}}{\alpha} >> 1$, i.e. we got only slightly worse results with $\alpha_{\text{pred}} = 0.2$ and $\alpha = 0.02$

check some of the details of the experiments there. However, as the code is not well-structured it was at times challenging to analyse. Furthermore, not all of the experiments are shared in the repository, for example there is no code which produces eigenspace experiments results or config for weight decay experiment.

The reproduced paper did not outlined self-contained description on the methods it used as it built upon previous works. Thanks to the detailed description of *BYOL* by Grill et. al. [2] we were able to reproduce the paper achieving similar results as the authors.

Due to time constraints we decided to use CIFAR-10 instead of STL-10 which was used in most of the experiments in the reproduced paper. However, claims tested by us in this work are not restricted to one dataset and we shown that they indeed hold in a different setting. One of the main challenges was the large amount of computations required for all the experiments, it took around 4 hours and 30 minutes to pre-train and fine tune a single model, and in total we trained for around 100+ hours.

Our work is implemented in TensorFlow and one of the challenges was differences between TensorFlow and PyTorch libraries. For instance, in PyTorch one of the parameters of the SGD optimizer is weight decay (L2 penalty), in TensorFlow we had to implement it by hand as TensorTlow's SGDW implements only Decoupled Weight Decay Regularization [17]. Furthermore, image augmentation methods such as ColorJitter from PyTorch do not have exact corresponding methods in Tensroflow. We used a custom way to do it so that augmentations are as close as possible to the original version.

# 7 Conclusion

In this work we study and reimplement three architectures used to give insight into self-supervised representation learning without contrastive pairs namely *BYOL*, *SimSiam*, *DirectPred* and their ablations. Our experimental results aligned well with both the theoretical analysis about the eigenspaces and the symmetric assumptions made in [1] and translate to other dataset than used in the paper. Lastly, we confirmed that SimSiam can be prevented from collapsing with the use of weight decay and adjusting a learning rate of predictor.

Furthermore, we confirm the claim that *DirectPred* outperforms its one-layer SGD alternative. However, we cannot report that *DirectPred* could outperform *Byol*. This may be due to the fact that we used CIFAR-10 as opposed to STL-10 in the original paper. This leaves us with the conclusion, that *DirectPred* gives valuable insights into the dynamics of unsupervised representation learning without contrastive pairs, but do not necessarily build new state of the art models themselves.

# 8 Ethical considerations

Self-supervised learning circumvents label scarcity which is one of the most common problems when applying ML to new scenarios. This can have both positive and negative consequences. On one hand, it can accelerate important developments for example in medical diagnosis. However, it can also be used in unethical ways such as in surveillance or military equipment. Furthermore, there will be less need for people labelling datasets which will result in reduction of job positions in this area.

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
