# OpenReview forum: "[Re] Understanding Self-Supervised Learning Dynamics without Contrastive Pairs "
_ML_Reproducibility_Challenge/2021/Fall — RC2021_

### Official Review · Reviewer_DFcw · 2022-03-07
**feedback**

**Rating:** 8
**Confidence:** 3

**Review:**

The report is well written and clearly explains the results obtained as well the insights the authors had from reproducing the work. Overall to me looks exactly as what I would have hoped for from a reproduction.

---

### Official Review · Reviewer_SWuk · 2022-03-19
**Marginally above threshold**

**Rating:** 6
**Confidence:** 4

**Review:**

1. The report contains a first page content for reproducibility summary, which has faithfully incorporated their major findings.
2. The scope is clearly and concisely stated.
3. The results are reproduced from scratch. Readable code and docs are submitted.
4. The reporters didn't contact the original authors.
5. Hyper-parameter search is not conducted.
6. Light ablation studies are conducted in the report.
7. The report shows that most of the reproduced results are consistent with the original paper (although some are reproduced on a new dataset). However,
        a). the reproduced accuracies are worse than the original paper by 0.51%~1.19%.
        b). the original results on the larger dataset and with larger training epochs are not validated in the report, due to time and resource limitation.
8. Concrete recommendations to the original authors for improving reproducibility are not given in the report.
9. The report is well organized and written, with minor typos.

---

### Meta-Review · Program_Chairs · 2022-04-09

**Recommendation:** Accept
**Confidence:** 5

**Metareview:**

The paper is well executed and accepted.

---

### Decision · Program_Chairs · 2022-04-09

**Decision:**

Accept

**Comment:**

Following the recommendation of reviewers and meta-reviewer, the paper is accepted for ML Reproducibility Challenge 2021, and will be published in the upcoming special edition of ReScience Journal.